# Development of Antimicrobial Stewardship Programmes in Low and Middle-Income Countries: A Mixed-Methods Study in Nigerian Hospitals

**DOI:** 10.3390/antibiotics9040204

**Published:** 2020-04-23

**Authors:** Eneyi E. Kpokiri, David G. Taylor, Felicity J. Smith

**Affiliations:** 1Department of Clinical Pharmacy and Pharmacy Practice, Faculty of Pharmacy, Niger Delta University, Bayelsa State 560103, Nigeria; 2Department of Practice and Policy, School of Pharmacy, University College London, London WC1N 1AX, UK; 3Faculty of Infectious and Tropical Diseases, London School of Hygiene and Tropical Medicine, London WC1E 7HT, UK

**Keywords:** antimicrobial resistance, stewardship programmes, low and middle-income countries

## Abstract

Antimicrobial resistance (AMR) is a major concern facing global health today, with the greatest impact in developing countries where the burden of infectious diseases is much higher. The inappropriate prescribing and use of antibiotics are contributory factors to increasing antibiotic resistance. Antimicrobial stewardship programmes (AMS) are implemented to optimise use and promote behavioural change in the use of antimicrobials. AMS programmes have been widely employed and proven to improve antibiotic use in many high-income settings. However, strategies to contain antimicrobial resistance have yet to be successfully implemented in low-resource settings. A recent toolkit for AMS in low- and middle-income countries by the World Health Organisation (WHO) recognizes the importance of local context in the development of AMS programmes. This study employed a bottom-up approach to identify important local determinants of antimicrobial prescribing practices in a low-middle income setting, to inform the development of a local AMS programme. Analysis of prescribing practices and interviews with prescribers highlighted priorities for AMS, which include increasing awareness of antibiotic resistance, development and maintenance of guidelines for antibiotic use, monitoring and surveillance of antibiotic use, ensuring the quality of low-cost generic medicines, and improved laboratory services. The application of an established theoretical model for behaviour change guided the development of specific proposals for AMS. Finally, in a consultation with stakeholders, the feasibility of the plan was explored along with strategies for its implementation. This project provides an example of the design, and proposal for implementation of an AMS plan to improve antibiotic use in hospitals in low-middle income settings.

## 1. Introduction

Rising AMR is viewed as a significant threat to global health. Concerted efforts to stem the rise in antimicrobial resistance and to ensure effective antimicrobial therapies for future generations is seen as a priority for the global community. Injudicious prescribing and use of antibiotics are considered principal drivers of increasing resistance. Thus, the central focus of initiatives to address AMR is on antimicrobial stewardship [1].

Whilst a global concern, inappropriate use of antibiotics and consequent problems of antimicrobial resistance are greater in low- and middle-income settings [2]. The determinants of antibiotic prescribing practices in resource-poor settings have been found to be wide-ranging [3]. Studies with health professionals have identified a poor appreciation of core principles, knowledge of antibiotic prescribing and problems of resistance, refs. [4,5,6] limited continuing medical education, ref. [7] a lack of updated policies and treatment guidelines, ref. [8] quality of antimicrobial medicines [9] and selective pressures from pharmaceutical companies [10]. There is commonly limited availability of expertise and diagnostic facilities to guide the choice of antibiotics [11,12].

Despite expansion of health insurance schemes, for many citizens, there is still no reliable funding stream to pay for care: consultation, diagnostic tests, and treatment. Dependence of many patients on out-of-pocket payments can affect prescribing decisions of prescribers and result in inappropriate and unregulated self-medication with antibiotics by patients [13,14]. To be effective, AMS strategies and interventions must be comprehensive, taking into account the different sectors in which antibiotics are used, promoting general hygiene, access to clean water, and public health measures. However, actions are required at all levels and this project focused on antibiotic prescribing in hospitals [3].

Suboptimal antibiotic prescribing practices remain common in many hospitals in low and middle-income countries (LMICs), and have been linked to increased incidence of antibiotic-resistant bacteria such as *Methicillin-Resistant Staphylococcus Aureus* (MRSA) [15]. Typical features of prescribing in LMICs include high volumes of antibiotic prescribing [16], although these higher levels can be partially linked to greater burden of infectious diseases [17]. Prescribing in LMICs has also been characterized by high empirical use of broad-spectrum antibiotics [8,18,19], non-availability of and/or poor compliance with treatment guidelines [20] and limited use of diagnostic tests [21]. 

Studies conducted in high-income countries have shown how antibiotic stewardship programmes can improve antibiotic prescribing in hospital settings, but these interventions have been poorly employed in LMICs [22] with only limited guidance regarding how this might be operationalized, in these settings [23]. The WHO practical toolkit for implementing AMS in LMICs provides guidance regarding strategies and procedures, also highlights the importance of exploring local context, priorities, and opportunities in the development of AMS programmes in LMICs [1]. In the development of a national strategic plan to address AMR in Thailand, participation of relevant stakeholders conferred pertinence to national and local priorities and challenges as well and stakeholders’ ownership to champion implementation [3]. Earlier AMS strategies to improve antibiotics use in LMICs include institutional policies, restrictions, and controlled usage using antibiotic charts [22,24]. A systematic review found that antibiotics stewardship programmes in LMICs had some positive impact on antibiotics use however studying the local context and adapting interventions will have more beneficial outcomes [23]. 

Engagement and consultation with local stakeholders will facilitate the identification of challenges and opportunities, enabling the development of an AMS programme that is likely to be effective in their local context [24,25]. This article reports the findings of a project, which employed a ‘bottom-up’ approach to identify important local determinants of antimicrobial prescribing practices and propose an AMS strategy and programme that would be pertinent and feasible in hospitals in a low-income setting leading to improvements in antibiotic prescribing practices.

## 2. Methods

### 2.1. Study Setting

This project was conducted in collaboration with hospitals in Bayelsa state, located in the Niger Delta Region of Nigeria. Data collection for this study was conducted from July 2015 to September 2017. Nigeria is classified as a developing nation in lower/middle income group [26]. In terms of life expectancy, maternal and child mortality and infectious disease burden, the situation in Nigeria is typical of other LMICs. Also, as for similar countries, resources for healthcare budgets are low (in Nigeria, less than 5% of the annual national budget is allocated for health) which is reflected in health service provision across the different units in the public healthcare sector [27].

### 2.2. Study Design

We employed a bottom-up mixed methods approach for this study. This involves distinct phases with different methods that are interlinked to explore and provide in-depth understanding of the subject. Antibiotic prescribing data was retrieved quantitatively and followed up with stakeholder interviews. The project was conducted in three stages. The first stage comprised a survey of antibiotic use across the selected study sites to establish the extent to which practices are typical of LMIC settings and a series of interviews with prescribers recruited across all hospitals, to identify important drivers of day-to-day prescribing practices as well as suggestions for interventions to improve antibiotic use. Stage 2 involved the application of an established theoretical model for behavioural change along with findings from stage 1, to develop a draft AMS strategy and plan. Stage 3 of the project comprised consultation with purposively selected stakeholders (a range of policymakers and health professionals with differing roles) to refine the proposals and inform an implementation plan.

### 2.3. Data Collection

A simple data collection tool (Appendix A) was developed to retrieve antibiotic prescribing data. Antibiotic prescribing data were obtained retrospectively from patients’ case notes selected by a simple randomized sampling. The patient appointment diaries in the medical records office was used to identify the case notes to be retrieved. The diary held a record of patients reporting to the hospital, the date of visit, department or unit patient was seen, the reason for the visit, diagnosis, and drugs prescribed. From the appointment diary, we compiled a list of case note numbers belonging to patients seen in the hospital between July and December 2015 who received one or more antibiotic prescriptions. Prescribing data collected was imported into IBM statistical package for social sciences (SPSS) version 22 and analysed using basic descriptive frequencies. Prescribers in the study hospitals were purposively selected to include diversity in terms of prescriber characteristics and setting. Participants were recruited from two tertiary (university teaching hospital and federal medical centre) and two secondary care hospitals (from mainland and island local government areas). Thus, together they provided coverage of tertiary and secondary, and smaller and larger institutions, urban and rural, and mainland and island settings across different levels of seniority. Prescribers details were obtained from the medical directory which contains a list of all medical doctors and dentists practicing in the region alongside their current practice facility, year of qualification and contact information. An information/ invitation letter (Appendix A) was developed and sent out to prescribers. Prescribers indicating interest to participate in the studies were provided a consent form (Appendix A) to sign just before the interviews. Participants were informed that all data will be anonymized prior to publication. All interviews were audio recorded and transcribed verbatim. Ethical approval was applied for and obtained from the ethical review committees of both tertiary hospitals and from the Bayelsa state management board prior to data collection (Appendix A).

## 3. Results


**Stage 1: Antibiotics prescribing practices and determinants**


### 3.1. Survey of Antibiotic Use in Hospitals in Bayelsa State, Nigeria

Across the four study sites, data relating to 809 patients prescribed one or more antibiotics were analysed. This comprised 264 (33%) and 246 (30%) from the two tertiary sites and 151 (19%) and 148 (18%) from the secondary care sites. More than half of the patients (53%) receiving antibiotics were ≤16 years old, and a further 21% aged 17–29 years. In 219 (27%) cases more than one different antibiotic was prescribed. The five most commonly prescribed antibiotics, making up 75% of prescriptions were broad-spectrum antibiotics including metronidazole, amoxicillin, amoxicillin and clavulanic acid, cefuroxime and ciprofloxacin suggesting a high rate of empirical prescribing. The majority (58%) of antibiotics were prescribed by brand, rather than generic name. Relevant laboratory tests such as microbial, culture and sensitivity tests to guide antibiotic treatment decisions were employed in only 15% of cases. In some cases where antibiotics were prescribed, tests such as abdominal scans and fasting blood sugar were carried out and did the presence of a bacterial infection was not indicated. There was limited observance (33% cases) of Nigerian Standard treatment Guidelines [28]. 

### 3.2. Interviews with Prescribers

To examine rationale for antibiotic prescribing in the local setting, interviews were conducted with 17 physicians (Table 1). Interviewees were purposively selected to include a diversity of physicians from across the 4 hospitals who prescribe antibiotics in their practice. The sample include male and female prescribers, with a spread from less than 5years up to more than 16 years in practice, different levels of seniority and specialty; and included an infection control lead and a medical director. Interview guides included predominantly open questions and principles of qualitative enquiry to explore antibiotic prescribing behaviours identified in stage 1. An initial coding frame was devised to enable a themed approach; constant comparison techniques were employed to facilitate more detailed analysis and explanation. 

Responses of participants revealed an awareness of, and concern about, suboptimal prescribing of antibiotics in the study settings, as found in stage 1. In their explanations of current practices, they highlighted high level system factors, local conditions, individual perceptions and practices all as potential determinants of antimicrobial use. These included, a lack of system support in local settings, especially inadequate laboratory services, excessive workload in the clinics; costs of drugs, especially in the light of patient socioeconomic status and prevalence of out-of-pocket payments; specific demands from patients, lack of availability of products, and shortcomings in training and knowledge and reliance on long-term prescribing habits, impact of pharmaceutical companies’ marketing and incentives. 

Respondents, notably, expressed regret that practices were often suboptimal, but saw the challenges as complex and often beyond their control. For example, whilst interviewees displayed positive views to the use of policies and guidelines, these were reported as being unavailable, out-of-date or difficult to comply with because of other system factors such as non-availability of drugs. Poor laboratory services, especially facilities that were non-functional, unstaffed and or open only for limited or irregular hours resulted in poor availability of diagnostic services and delays in obtaining results of sensitivity tests. This was seen by many as contributing to high levels of empirical prescribing of broad-spectrum antibiotics. Patients without health insurance, necessitating out-of-pocket payments, could also be a barrier to requesting diagnostic tests.

Thus, cost and affordability for patients were often cited by prescribers who reported compromising in the choice of drug to cheaper alternatives to ensure some antibiotic cover. The range of products procured and stocked in the hospitals also influenced drug choice. The negative impact of pharmaceutical companies promoting brands and incentives for prescribing was also raised. Concerns about quality and effectiveness of some non-branded products was also reported as a driver for brand-name rather than generic prescribing. 

Whilst, the patterns and determinants of antibiotic prescribing in these hospitals were often typical of those reported in other LMIC settings, the observations and experiences of prescribers, facilitated their engagement in proposing suggestions and recommendations for AMS interventions pertinent to their settings and challenges. Prescribers identified a diversity of macro and micro-level factors as determinants of antibiotic use and potential targets of an AMS programme. These are summarized in Table 2.


**Stage 2: Development of an AMS plan using a framework of behaviour change**


To guide the development of an AMS programme, a validated framework the *Behaviour Change Wheel* (BCW) was employed [29]. Theoretical frameworks have been commonly applied to facilitate the development of complex interventions [30,31]. The BCW was selected because it provides a comprehensive, systematic approach incorporating high-level system factors, local context, and individual characteristics, all of which, in stage 1, were found to be determinants of prescribing practices in the local settings. In addition to seven policy categories (the outer wheel) the BCW also presents a methodology for intervention on the basis of function or approach. These intervention functions also coincided with suggestions from prescribers and could be incorporated into AMS programmes and strategies. The inner core of the BCW focuses on behaviour change at the individual level. It applies the concepts of capability, opportunity and motivation to examine prescribers’ roles, experiences, and perspectives. Through the application of intervention functions, these are seen as a central focus for behaviour change. (See Figure 1).

Using the BCW framework (policy categories and intervention functions) along with the domains and specific suggestions of prescribers and stakeholders, a draft framework for an AMS programme was constructed (Table 3). In this draft framework, the suggestions made by prescribers were also identified as those that would be part of a short/immediate or longer-term strategy. This was viewed as important, as this setting (as other LMICs) interventions are more likely to be constrained because of limited resources and infrastructure.

Some interventions could be implemented immediately, in that the available resources could support the initial start-up, and that existing committees (e.g. Drugs and Therapeutics Committee -DTC) could undertake planning and execution. These interventions may include increasing local awareness, development of guidelines and protocols, and implementation of local audit and monitoring. Other measures would require policy development and planning, at a higher (sometimes Government) level, and thus will be goals achievable as part of a longer-term strategy, e.g. allocation of additional finance, continued roll-out of health insurance to ensure affordability for patients, enhanced human resources.

The final consideration in the development of the plan for AMS involved the sequencing of interventions. This will depend on contexts and priorities in different hospitals, e.g. some hospitals may be setting up a programme for the first time, others updating an existing strategy. In practice, in all sites, continual review will be required. 


**Stage 3: Consultation with key stakeholders**


The third stage of the project was consultation with key stakeholders. The aim of this was to examine the feasibility and acceptability of recommendations from the perspectives of different stakeholders and to inform a plan for implementation relevant to, and workable in, their local settings. 

A purposive sample including key members of the hospital, clinical and administrative staff and other relevant stakeholders in regulatory/policy settings were recruited. Eight participants were drawn from different areas of practice and included a hospital medical director, prescribers, pharmacists, microbiologist, long-standing member of Drugs and Therapeutic Committee (DTC), chairperson of the medical association and representative from the State Ministry of Health. In this consultation, data were collected in one to one interviews. The stakeholders expressed their views on priorities for AMS, opportunities, and challenges for implementation in their settings.

## 4. Discussion

### 4.1. Feasibility and Priorities for Implementation

The draft framework for implementing AMS in the local settings was presented to the stakeholders and this was largely viewed as apposite and feasible. Their top priorities for implementation include increased awareness and education about antibiotic resistance, development, and provision of policies and guidelines on antibiotic use, monitoring and surveillance of antibiotic use, improved laboratory and diagnostic services and ensuring availability and quality of products.

Some of these interventions had already been initiated, particularly in the teaching hospital in accordance with requirements for accreditation. In relation to these priorities, stakeholders’ perspectives regarding the opportunities and challenges in their settings and potential action plans were explored. The findings are reported in Table 4.

### 4.2. Set up and Operation of the AMS Plan

The proposed AMS plan encompasses the policy categories, intervention functions and sources of behaviour change as outlined in the three layers of the BCW in that it identifies the interacting roles of the government (in terms of wider health policy), the institutions (the hospitals) and individuals (practices and behaviours of prescribers, healthcare providers and patients. In the short term, the focus is on the institutional level, in particular, interventions that address the local context and initiatives that will support and effect behaviour change in the prescribing and use of antibiotics.

In the consultations with stakeholders, there were common strategies that emerged as relevant to the different priorities and components of the AMS plan. First and foremost is to build the AMS team. The make-up and operation of the team should meet the needs of all elements of the AMS programme. In the first instance, a high-profile leader should be identified to champion the programme and give prominence and credibility to the programme and to encourage wide engagement. Involvement across professional groups, roles, and levels of seniority will facilitate a collaborative approach in which the perspectives and interests of all stakeholders can be represented. Thus, other members will include staff from the medical team, pharmacy, nursing, microbiology, medical records and hospital management/administration. The team also needs to include individuals with more specialist expertise to guide and implement particular aspects of the programme, e.g. laboratory skills or experience in procurement, interactions with outside policy bodies or companies. The programme will also require the involvement of the people with experience in publicity, organization of events and administrative skills.

A lack of sufficient specialist expertise was highlighted in relation to a number of the priority areas, e.g. with regard to leading local surveillance and research or laboratory skills. A first step here may be to identify existing relevant expertise across the relevant professional groups.

In some cases, there may be staff in-post who, with some additional training, would be able to undertake some AMS activities alongside their current roles. A longer-term plan may require further training and/or recruitment of specialist expertise.

Through regular meetings (suggested by stakeholders as monthly in the first instance and perhaps bi-monthly once AMS activities become established) the team will be responsible for developing a mission statement. Subgroups should be set up to take forward specific tasks and objectives set by the wider team. A mechanism will be required for maintaining and distributing local guidelines throughout the hospital to reach and engage relevant staff. Other activities of the AMS team will include regular review of antibiotics use, documentation of local bacterial sensitivity patterns and revisions of antibiotic formulary lists. The team will also plan for surveillance and auditing of practice (prescribing, dispensing and laboratory procedures), document findings and generate appropriate reports. These reports will be reviewed and presented to staff as part of communication and education, highlighting good practices as well as areas requiring improvement.

## 5. Conclusions

The findings from this project have informed a plan for antimicrobial stewardship in a low-resource setting. Whilst this work commenced before the publication of the WHO toolkit, it provides a road map to developing an AMS programme in an LMIC with locally retrieved data. This project took a bottom-up approach combined with application of a theoretical framework of behaviour change, leading to the development of a multifaceted AMS plan. While the AMS plan developed here has been designed to be implemented in a specific LMIC setting, early stages of the project confirmed that many of the practices regarding antibiotic use, prescribing patterns, barriers, and opportunities for change are similar to those in other LMICs. Therefore, the findings presented here can be implemented in similar resource-limited settings.

## Figures and Tables

**Figure 1 antibiotics-09-00204-f001:**
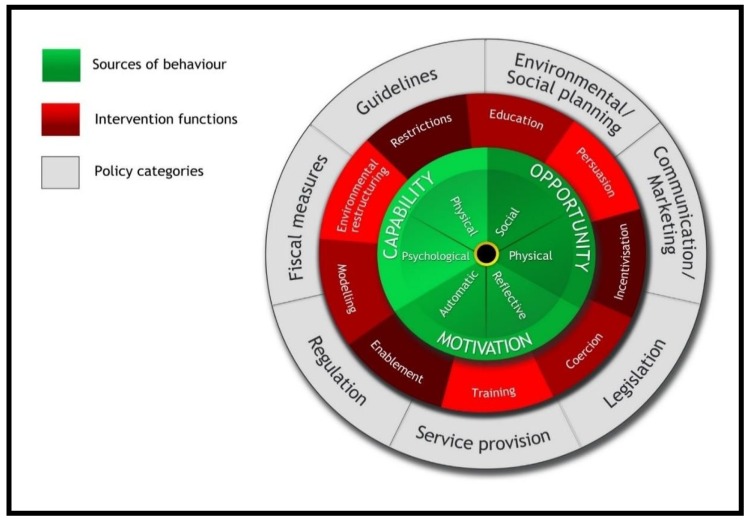
The Behaviour Change Wheel [29].

**Table 1 antibiotics-09-00204-t001:** Characteristics of participants.

Characteristics		Number of Participants n = 17 (%)
Gender	Male	12(70)
	Female	5 (30)
Years of practice	≤5	7 (40)
	6–10	4 (24)
	11–15	4 (24)
	≥16	2 (12)
Rank	Medical officer	7 (40)
	Senior medical officer	3(18)
	Principal medical officer	4 (24)
	Consultant	3 (18)
Specialty	Internal Medicine	6 (35)
	Paediatrics	5 (30)
	Surgery	3 (18)
	Obstetrics & Gynaecology	2 (12)
Level of care	Tertiary	11(65)
	Secondary	6 (35)

**Table 2 antibiotics-09-00204-t002:** Domains for antimicrobial stewardship programmes (AMS) actions as identified by prescribers.

Improve diagnostic testing and services
Education and training for prescribers Enhanced local awareness campaigns and advertising
Provide and maintain up-to-date local policy and prescribing guidelines Enforcement of local guidelines Restrictive prescribing for some antibiotics (in accordance with local guidelines) Regulation of Pharmaceutical Company marketing
Conduct routine audits and local studies to inform guidelines, identify problems and monitor practice
Steps to improve the availability of drugs: Ensure continuous supplies (with a separate focus on local prescribing guidelines and specialist needs) Improve affordability for patients
Assurances of the quality of generic products
Enhanced role for pharmacist interventions

**Table 3 antibiotics-09-00204-t003:** Draft framework for AMS showing application of the Behaviour Change Wheel (BCW) policy categories, intervention functions, and specific recommendations.

Policy Category	Intervention Functions	Suggestions and Recommendations from Stages 1 and 2	Implementation Category
Communication/marketing	Education Persuasion Training	Increase awareness of problems and rational prescribing Use of fliers and posters as well as on-line and face-to-face methods. Education and training via short courses, workshops, use of local data to educate prescribers, highlight best practice and address problems	ST
Guidelines	Persuasion Restriction Modelling Enablement	Provision of up-to-date guidelines and treatment protocols Enablement, and models, of best practice Strict protocols to restrict access to reserved products, including restricted pharmacy dispensing	ST MT
Fiscal	Environmental restructuring Enablement	Improvements to lab facilities to provide enabling environment for rational prescribing. Wider coverage of health insurance, Stocking of low-cost generics to promote affordability for patients Invest in quality assurance units to safeguard product quality, so quality is not a barrier to optimal prescribing	LT
Regulation	Training, Restriction, Coercion Enablement	Implementation of prescribing guidelines Regular auditing of practice to regulate and inform improvements to practice Collection and use of local data to provide directly relevant feedback to practitioners and teams	ST S/MT
Legislation	Environmental restructuring Restriction, Enablement	Support at Government/health policy levels for prioritization of structural changes, and wider enforcement.	LT
Environmental/social planning	Environmental restructuring Enablement	At Government/health policy and institutional levels: interventions as above to ensure enabling environment	M/LT
Service provision	Education Training Restriction Environmental restructuring Enablement	Implementation of guidelines Availability and use of laboratory facilities Regular auditing of practice to identify and address problems; and ensure continued enabling environment	MT/LT S/M/LT

ST- Short term, MT-Medium term, LT- Long term.

**Table 4 antibiotics-09-00204-t004:** Priorities, opportunities, and barriers leading to action plan.

*Increasing awareness and education about antibiotic resistance* It was recommended that initiatives to increase awareness should target patients, other health professionals as well as prescribers. Stakeholders perceived that there was a general awareness, and one hospital reported to have already held a presentation. Thus, there were some opportunities to build on existing initiatives. Stakeholders saw important challenges in addressing attitudes and behaviours. Securing engagement and attendance and promoting a desire among practitioners to change was highlighted. Recommendations for AMS action plan and implementation: Appoint high profile leader/ AMS champion to act as a focal point to encourage engagement and oversee publicity campaigns and training Employ a wide-ranging approach to increasing awareness and training appropriate to all stakeholders including patients.
*Provision of policies and guidelines* Stakeholders reported that in the past prescribing guidelines had been developed, but these were commonly seen as not up-to-date or not readily accessible. A potential challenge in the development of guidelines was having sufficient expertise for their development. However, it was acknowledged that a wide range of stakeholders e.g. DTC, infectious disease clinicians and scientists, pharmacists would bring together their professional expertise. Ensuring engagement, ownership and acceptance by all stakeholders was seen as important for co-operation, compliance and enforcement. In one setting, restricted dispensing had already been accepted by the DTC, which included a procedure for review and approval of restricted products. Acceptance and implementation may be facilitated by a collaborative approach (scientists and health professionals) to their development. An inter-professional approach to development of policies and guidelines bringing together and enhancing local expertise, ensuring local relevance and ownership. A collaborative approach to maintenance and review of guidelines and to agree oversight and co-operation in their enforcement.
*Monitoring and surveillance of antibiotic use* On-going monitoring of the use of antibiotics and local research on infections and resistance could inform more rational use and was also of value in the development of policies and guidelines. A monitoring or surveillance programme was also seen as a way of engaging professional groups and bringing them together in a shared AMS programme. In one site discussions had already begun. The principal barrier identified was having sufficient personnel with expertise to lead for an on-going surveillance and research programme. Possible action points were: Develop programme for on-going monitoring which engages different professional groups Identify individuals and address any training needs in audit, surveillance, and research
*Improved laboratory and diagnostic services* Improvement of laboratory services and training of scientists to reduce empirical prescribing was identified as a requirement to guide judicious antibiotic prescribing. The key challenge highlighted was the recruitment of scientists with sufficient expertise. It was suggested that pharmaceutical companies may be able to assist, e.g. with the provision of sensitivity discs for their products. Thus, as part of an action plan: AMS team could identify steps that could be taken, in both short and medium term, towards the enhancement of laboratory facilities, services and expertise.
*Procurement and quality assurance* Steps to ensure continuity in availability, affordability and trust in the quality of products (especially low-cost generics) was seen as essential for the successful operation of an AMS programme in short, medium, and longer term. Barriers to access to quality medicines was seen as encompassing manufacturing and regulation, affordability for patients and prescribing practices. In terms of quality assurance, facilities, personnel, and expertise was highlighted as a challenge. Possible collaboration between manufacturing and regulatory bodies was also mentioned. Despite the challenges, it was viewed that the AMS team should: Identify potential opportunities to improve procedures for quality assurance, availability, and affordability of products

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
