# Peer review of "Development of Antimicrobial Stewardship Programmes in Low and Middle-Income Countries: A Mixed-Methods Study in Nigerian Hospitals"

_antibiotics, 2020, doi:10.3390/antibiotics9040204_

Round 1

Reviewer 1 Report

Dear authors:

Thank you for the opportunity to review this manuscript. It is a challenging and yet important task to implement antimicrobial stewardship, especially in the resource-limited setting. The authors should be commended for their work in Nigeria and for engaging practitioners on the ground as well as stakeholders in the development of antimicrobial stewardship activities. However, the study methodology should be elaborated upon in detail to improve its scientific rigor. Additionally, the authors should also highlight important learning points from their own experiences such that the manuscript is also beneficial for others who are implementing antimicrobial stewardship in resource-limited settings.

Introduction

  • Studies that reported the development and results of antimicrobial stewardship programs in resource-limited settings should be discussed.

General

  • While not the focus of my review, I believe that the manuscript in general may be improved in sentence structure and punctuation. Abbreviations should also be defined when they first appear in the manuscript.

Methods

  • Please define study time frame.
  • Additionally, how were patients or antibiotic prescriptions identified for review? How were the physicians selected for interview?
  • Was institutional review board approval obtained and did the physicians provide consent for their information to be shared in this publication?

Results

  • Section 3.1: Please define the more common “broad-spectrum” antibiotics used. What were the “relevant laboratory tests” used?
  • Section 3.2: Please provide more details on the 17 physicians, e.g. specialty area, site of practice, etc.

Author Response

Thank you for reviewing our manuscript and providing such positive feedback regarding the current relevance and potential value of the paper.

INTRODUCTION: studies that discussed antimicrobials stewardship programmes in other LMICs have been discussed in pages 3-4 (references 22-25).

GENERAL: Overall sentence structure has been revised, entire manuscript read through and English checked. All abbreviations are now defined at first use.

METHODS- Study time frame has been defined on page 4. Recruitment of patients and antibiotic prescribing records is now included on page 5. Relevant ethical applications were made, and approvals obtained for this study. Participants consent was also obtained to share findings anonymously. This is reported on pages 5-6 of the manuscript.

RESULTS – List of broad-spectrum antibiotics commonly prescribed has been specified on page 6. Relevant tests for antibiotic prescribing clarified on page 6. Additional details of prescribers as requested now included in Table 1 (page 7).

Reviewer 2 Report

This manuscript is, " Towards implementing antimicrobial stewardship programmes in low- and middle-income countries: A mixed-methods study in Nigerian hospitals " by Dr. Kpokiri and his/her colleagues. This study is a mixed methods conducted in three stages. a series of interviews with prescribers recruited across all hospitals; the application of an established theoretical model for behavioral change; and consultation with purposively selected stakeholders.

In my opinion, this manuscript seems to be a political annunciation, because the data and analyses are lacking in their work. The mixed method is novel, but there is no information to support the ASP is clinically beneficial in these countries.

Author Response

Thank you for taking time to review our manuscript. Antibiotic Stewardship is a health policy priority and therefore should be on the political agenda of all governments, especially in low-income countries where the challenges are greater. 

Detailed analyses were undertaken for all parts of the study. Steps were taken to ensure they were scientifically robust across all data sets and the findings of each informing subsequent stages and overall aims of the work. 

Mixed methods approaches are now commonly employed in health research and policy development as they enable important issues to be examined and addressed, taking into account different perspectives. 

As recognised by the WHO in the recent tool kit published for ASP for low-income settings have to be relevant to the macro and micro context in those settings. As a consequence of the ‘bottom-up’ and mixed methods approach this project provides an example of development of clinically beneficial ASP in a low-income setting. 

Round 2

Reviewer 1 Report

Thank you for the opportunity to review this manuscript. It is a challenging and yet important task to implement antimicrobial stewardship, especially in the resource-limited setting. The authors should be commended for their work in Nigeria and for engaging practitioners on the ground as well as stakeholders in the development of antimicrobial stewardship activities. The authors addressed my previous concerns in their revisions. Minor typographical errors still exist which I hope will be corrected in the editing process. I have no further concerns for the manuscript to be published.

Author Response

Thank you to the reviewer for reading the paper and for the comments. We have reviewed the paper to correct a few typographical errors as suggested. We acknowledge that the reviewer now views the manuscript suitable for publication.

Reviewer 2 Report

The present study indicated this project provides an example of the design, and proposal for implementation of an AMS plan to improve antibiotic use. However, any data to present the improved in antimicrobial administration after the AMS implementation was lacking in the revised manuscript. Only the structure and step to establish the ASM was demonstrated in your work.

Author Response

Thank you for your comments. The purpose of this project was to inform the development of an AMS programme taking into account the importance of local contexts, settings, priorities, and drivers of prescribing practices as recommended in the recent toolkit of the WHO with regard to the development of AMS in low-income countries. Thus, the goal was not the implementation of AMS but to develop a proposal to support the implementation of AMS in low-income settings.

However, to respond to this point and prevent any confusion regarding the purpose of the paper, we have modified the title of the paper from:
'Towards implementing antimicrobial stewardship programmes in low- and middle-income countries: A mixed-methods study in Nigerian hospitals'
to:
'Development of antimicrobial stewardship programmes in low- and middle-income countries: A mixed-methods study in Nigerian hospitals'

We have reviewed the manuscript to identify and correct any errors.

Round 3

Reviewer 2 Report

I agree the modification in your revised manuscript. This is an fully informative study. The reviewers' comments has been well reflected. Thank you.